# The Prevalence of Small for Gestational Age and Extrauterine Growth Restriction among Extremely and Very Preterm Neonates, Using Different Growth Curves, and Its Association with Clinical and Nutritional Factors

**DOI:** 10.3390/nu15153290

**Published:** 2023-07-25

**Authors:** Ioanna Kakatsaki, Styliani Papanikolaou, Theano Roumeliotaki, Nicolina Hilda Anagnostatou, Ioanna Lygerou, Eleftheria Hatzidaki

**Affiliations:** 1Neonatal Intensive Care Unit, Department of Neonatology, University General Hospital of Heraklion, 71500 Crete, Greece; joanne_k1991@hotmail.com (I.K.); stylpapa12@gmail.com (S.P.); nicolehilda@gmail.com (N.H.A.); lygerouped@gmail.com (I.L.); 2Clinic of Preventive Medicine and Nutrition, Division of Social Medicine, School of Medicine, University of Crete, 70013 Crete, Greece; roumeliot@uoc.gr; 3Neonatology, School of Medicine, University of Crete, 70013 Crete, Greece

**Keywords:** prematurity, restricted growth, EUGR, SGA, extremely preterm, very preterm, INTERGROWTH-21st, Fenton2013

## Abstract

Monitoring the growth of neonates in the Neonatal Intensive Care Unit (NICU) using growth charts constitutes an essential part of preterm infant care. Preterm infants are at increased risk for extrauterine growth restriction (EUGR) due to increased energy needs and clinical complications. This retrospective study compares the prevalence of small for gestational age (SGA) at birth and EUGR at discharge in extremely and very preterm neonates hospitalized in the NICU of a tertiary hospital in Greece, using different growth curves, and it examines the associated nutritional and clinical factors. Fenton2013 and INTERGROWTH-21st growth curves were used to calculate z-scores of birth weight (BW) and weight, length, and head circumference at discharge. The study includes 462 newborns with a mean BW of 1341.5 g and mean GA of 29.6 weeks. At birth, 6.3% of neonates were classified as SGA based on Fenton2013 curves compared to 9.3% with INTERGROWTH-21st growth curves. At discharge, 45.9% of neonates were characterized as having EUGR based on the Fenton2013 weight curves and 29.2% were characterized based οn INTERGROWTH-21st curves. Nutritional factors such as the day of initiation, attainment of full enteral feeding, and the duration of parenteral nutrition were associated with EUGR by both curves. The prevalence of SGA and EUGR neonates differs between the two growth references. This shows that further evaluation of these charts is needed to determine the most appropriate way to monitor infant growth.

## 1. Introduction

Assessing and monitoring the postnatal growth of prematurely born infants is fundamental to Neonatal Intensive Care Units (NICUs). However, there is no standardized approach among neonatologists regarding how the postnatal growth of preterm infants should be monitored, and the optimum pattern of growth has yet to be determined [1,2].

Growth curves serve as monitoring tools for assessing the growth of preterm infants at birth and in the postnatal period. In clinical practice, two widely utilized growth charts are the Fenton 2013 and the INTERGROWTH-21st growth charts [3]. The 2013 Fenton growth charts are one of the most commonly used reference charts and were created based on the theory that the growth of a preterm infant should follow that of a normal human fetus [4]. The use of these growth charts has been questioned in recent years since the growth of a fetus in utero and the growth of an infant ex utero are different biological processes due to environmental and nutritional factors [5]. Later developed growth monitoring tools, such as the INTERGROWTH-21st preterm postnatal growth standards, are prospectively constructed standard curves based on a subpopulation of preterm infants born from healthy mothers with no related complications in pregnancy and no evidence of fetal growth restriction [6]

Feeding practices in NICUs have exhibited significant variation, particularly in previous times [7]. Traditionally, a conservative feeding approach involved fasting shortly after birth, introducing feeds after four days, and gradually increasing feeding volume at a maximum rate of 24 mL/kg/day. This approach aimed to minimize the risks of necrotizing enterocolitis (NEC) and early onset sepsis [8,9]. Over time, a more liberal approach was followed, which involves early trophic feeding within 24–48 h after birth and gradual advancement in an effort to achieve full enteral feed in a shorter period [10].

Preterm infants are at increased risk for extrauterine growth restriction (EUGR) due to often being born small for gestational age (SGA), increased energy needs and prematurity-associated morbidities [11]. The identification of EUGR and SGA neonates is crucial as it carries significant implications to their health in the neonatal period and later in life [12,13,14]. Nonetheless, the prevalence of SGA and EUGR exhibits considerable variation across studies, which is closely related to the choice of a specific growth chart for assessment [15].

The aim of this study was to compare the prevalence of SGA and EUGR in infants born before the 32nd week of gestation, using the Fenton2013 and INTERGROWTH-21st growth reference curves, and determine the nutritional and clinical factors associated with restricted postnatal growth by both charts.

## 2. Materials and Methods

### 2.1. Study Setting and Population

This is a retrospective study conducted at the Department of Neonatology/Neonatal Intensive Care Unit of the University General Hospital of Heraklion (PAGNI), a tertiary referral hospital in Crete, Greece, with nearly 600 admissions annually. Preterm infants born before 32 weeks admitted to the NICU from January 2008 to December 2022, who survived until discharge to home, were considered eligible participants for the study. Exclusion criteria were major congenital malformations and genetic syndromes. The final study population consists of 462 extremely and very preterm infants. Ethical approval was obtained from the Scientific Council of the Hospital (protocol number 23000).

### 2.2. Indicators of Growth and Development

Birth weight, gestational age, and child’s biological sex were recorded by hospital staff within 1 h after delivery. Weight, length, and head circumference (HC) were measured at discharge following standard clinical protocols. Birth and discharge weight were measured using adapted patient incubator electronic and digital infant weighing scales.

Gestational age (GA) was determined according to the last menstrual period, an early prenatal ultrasound or calculated directly in case of in vitro fertilization. Small for gestational age (SGA) was defined when birth weight was below the 10th sex-specific percentile for GA. Postmenstrual age (PMA) at discharge was calculated as gestational age plus chronological age in weeks [16]. Extrauterine growth restriction (EUGR) was defined as weight, length, or HC at discharge below the 10th sex- and postmenstrual age-specific percentile. Percentile and z-scores of weight at birth were calculated using both the Fenton growth charts and the INTERGROWTH-21st very preterm size charts [4,17]. Size at discharge was also calculated using the Fenton charts and INTERGROWTH-21st postnatal growth standards for preterm infants [4,18].

### 2.3. Nutritional Data 

Assessed nutritional factors included in our study were the day of life that enteral and parenteral nutrition were initiated, the time to reach full feeds and the duration of parenteral nutrition. It should be noted that in the studied period of 15 years, nutrition protocols have changed, adapting to current international recommendations regarding enteral and parenteral feeding [19,20,21,22,23,24,25,26]. During the recent years, total parenteral nutrition (TPN) was initiated on the first day of life, and enteral trophic feeding started as soon as possible, either with expressed own mother’s milk, when available, or with formula designed for premature infants [27]. Enteral feed volumes were gradually increased on a daily basis for each infant, provided that they tolerated the feeding, while simultaneously reducing the amount of parenteral nutrition [28].

### 2.4. Clinical Data

Additional information recorded for each infant included multiple gestation, duration of hospitalization (days), and morbidity data such as pneumothorax [29], bronchopulmonary dysplasia (BPD), early (<72 h) and late (≥72 h) suspected and culture-proven sepsis, anemia requiring at least one blood transfusion [30], necrotizing enterocolitis (NEC) defined according to modified Bell’s criteria (stage IΙIB) [31], hemodynamically significant patent ductus arteriosus (hsPDA) determined by a combination of echocardiographic and clinical criteria and requiring pharmacological treatment [32], retinopathy of prematurity (ROP) (≥Stage II) [33,34], cystic periventricular leukomalacia (PVL) and intraventricular hemorrhage (IVH) grade I–IV based on ultrasound diagnosis [35,36]. Finally, the duration in days of respiratory support, mechanical ventilation, non-invasive ventilation, and oxygen administration was recorded.

Different criteria were used to define some of the above morbidities over the years of the study. BPD until 2018 was defined as the need for supplemental oxygen for 28 consecutive days or supplemental oxygen at 36 weeks PMA; after 2019, according to Jensen et al., the definition of BPD was modified and included respiratory support plus oxygen versus oxygen only in order to classify BPD at 36 weeks PMA [37]. Sepsis is a clinical syndrome in a neonate characterized by a combination of clinical, biochemical, and microbiological data. In our study, both suspected with negative blood culture and confirmed by blood culture sepsis were assessed [38,39]. 

### 2.5. Statistical Analysis

Descriptive statistics are expressed as mean and standard deviation for continuous variables and as frequencies and percentages for binary or categorical data. The paired *t*-test was used to compare sex-specific z-scores at birth and discharge between the two growth references (Fenton2013 and INTERGROWTH-21st). Cochran’s Q test was used to compare the prevalence of SGA and EUGR in the study population based on the different growth references. Bivariate comparisons of nutritional and clinical factors between the EUGR and non-EUGR groups previously defined were conducted using the parametric Student’s *t*-test or non-parametric Mann–Whitney for continuous variables, and Fisher’s exact test was used for categorical variables.

We used multivariate linear regression models to explore the association of sex-specific z-scores of weight, length, and HC at discharge with nutritional and clinical factors. Estimations are presented regarding β coefficients and their 95% confidence intervals (CI). For the binary outcomes of EUGR, logistic regression models were applied, and estimations are presented in terms of odds ratio (OR) and their 95% confidence interval (CI). Possible factors identified from the univariate comparisons were entered into the regression models for the multivariate models. Due to highly correlated factors, some variables were excluded according to the correlation matrix and the variance inflation factor (VIF), and only one of the highly correlated variables was included in the model. Additional sensitivity analysis was performed by stratifying admissions before and after 2018 in order to examine possible modification due to the change in nutritional feeding practices during the course of the study.

All hypothesis testing was conducted, assuming a 0.05 significance level and a 2-sided alternative hypothesis. All statistical analyses were performed using Stata Software, version 13 (Stata Corp LP, College Station, TX, USA).

## 3. Results

### 3.1. Infant Characteristics

Perinatal, clinical, and nutritional characteristics of the study population are presented in Table 1. The mean (±SD) gestational age was 29.6 (±1.7) weeks, and 17.5% of infants were born extremely preterm (<28 weeks). Newborns had a mean BW of 1341.5 (±363.3) grams, and 55.8% were male. The mean duration of hospitalization was 54 (±25.8) days. At discharge, the mean PMA was 37.3 (±2.9) weeks, and the mean weight was 2415.9 (±406.7) grams. Nutritional data indicate that parenteral feeding was introduced the first day, with a median (IQR) duration of 9 (5–18) days. Enteral feeding starting age was 4 (2–6) days, and full enteral feeding was achieved at 13 (9–22) days. The most common morbidities among preterm infants were anemia (45.7%) and late-onset sepsis (26.4%). BPD and ROP were present at 14.3% and 6.7% of cases, respectively. Mean (±SD) duration of invasive mechanical ventilation was 3.3 (±7.6) days, and non-invasive ventilation was 9.7 (±13.2) days, whereas oxygen administration was used 11.6 (±18.3) days. 

In the course of years of the study, nutrition protocols strain to adapt to international recommendations regarding enteral and parenteral feeding, and this change is depicted in Figure 1. A significant decline is observed after year 2018 (*p* < 0.001) for the median day of enteral feeding initiation (median (IQR) before 2018: 5 (3–7) days; after 2018: 2 (2–4) days) and full enteral feeding achieved (before 2018: 14 (9–24) days; after 2018: 11 (8–17) days). 

### 3.2. The Fenton2013 and INTERGROWTH-21st Growth References

Infant size at birth and at discharge was examined by applying the Fenton2013 and INTERGROWTH-21st growth references, as previously described, and the results are presented in Table 2 and Figure 2. The prevalence of SGA infants at birth is 6.3% based on Fenton2013 curves and 9.3% based on INTERGROWTH-21st very preterm size at birth reference charts. Extrauterine growth, assessed by means of weight, length, and head circumference measured at discharge, was found to be significantly different between the two growth standards. The Fenton2013 growth curves produced a higher prevalence of EUGR compared to INTERGROWTH-21st for the weight (45.9% and 29.2%, respectively) and length (36% and 34.4%, respectively). Regarding HC, our study revealed a higher prevalence of EUGR when the INTERGROWTH-21st postnatal growth charts were used.

### 3.3. Nutritional and Clinical Factors Associated with EUGR

Bivariate comparisons between perinatal, nutritional, and clinical information with EUGR (Table 3) showed that postnatal growth failure was more prevalent in extremely preterm infants (by Fenton, *p* = 0.003 and by INTERGROWTH-21st, *p* < 0.001) and those who had stayed longer in the NICU (*p* < 0.001, by both reference curves). Clinical conditions associated with restricted postnatal growth by both growth references were BPD, late-onset sepsis, anemia, hsPDA and ROP. In addition, EUGR infants needed more days of respiratory support, mechanical or non-invasive ventilation and oxygen supply.

All three nutritional factors accounted for were significantly related to postnatal growth failure with parenteral supported feeding lasting longer and full enteral feeding being achieved later by at least one week (*p* < 0.001). Further analysis revealed a significant decrease in the prevalence of EUGR after the year 2018 when nutritional practices altered toward international standards compared to previous years (36.6% compared to 51.2% by Fenton2013 (*p* = 0.003); 21.3% compared to 33.7% by INTERGROWTH-21st (*p* = 0.005)).

### 3.4. Multivariate Analysis

In the multivariate analysis presented in Table 4, 1 SD change of weight at discharge was increased for extreme prematurity by 0.53 (95% CI: 0.23 to 0.84) and 0.48 (95% CI: 0.17 to 0.80) based on the Fenton2013 and INTERGROWTH-21st growth references, respectively, but no effect was observed on the prevalence of EUGR. Birth weight was positively associated with weight at discharge and had a protective effect on EUGR (OR: 0.21 (95% CI: 0.14 to 0.31) by Fenton2013; OR: 0.27 (95% CI: 0.19 to 0.39) by INTERGROWTH-21st). On the contrary, long-term hospitalization was a burdensome factor for decreased weight at discharge, which was expressed in SD change by −0.04 (−95% CI: 0.04 to −0.03) and risk of EUGR (OR: 1.05 (95% CI: 1.03 to 1.07).

Weight at discharge SD score was inversely associated with the number of days needed to achieve full enteral feeding (EN), and the duration of parenteral feeding (PN) increased infant size. The day of EN initiation was a risk factor for EUGR by INTERGROWTH-21st reference (OR: 1.16 (95% CI: 1.05 to 1.25), whereas the risk of EUGR by Fenton2013 reference was higher by 7% for each day that EN was delayed (OR: 1.07 (95% CI: 1.03 to 1.12).

Lastly, the clinical conditions that retained significance in the multivariate models were anemia (β: 0.33 (95% CI: 0.10 to 0.56) and β: 0.31 (95% CI: 0.06 to 0.55)) and hsPDA (β: −0.41 (95% CI: −0.75 to −0.07) and β: −0.39 (95% CI: −0.75 to −0.04)) for Fenton2013 and INTERGROWTH-21st weight SD change, respectively. BPD was associated only with weight SD change based on INTERGROWTH-21st curves (β: 0.37 (95% CI: 0.00 to 0.73)). The risk of EUGR based on Fenton2013 curves was inversely related to anemia (OR: 0.49 (95% CI: 0.25 to 0.95)), whereas based on INTERGROWTH-21st curves, it was tripled for infants developing late-onset sepsis (OR: 3.08 (95% CI: 1.56 to 6.08)).

Sensitivity analysis by data stratification according to the year of admission (before or after 2018) showed that the perinatal factors accounted for remained highly significant for both growth references for weight z-scores (Appendix A). Similar effects were observed for the nutritional and clinical factors, even if significance was not attained for all predictors, which was possibly due to sample size reduction. The prevalence of EUGR was lower for admissions after 2018 by both Fenton2013 (51.2% vs. 36.6%) and INTERGROWTH-21st (33.7% vs. 21.3%) weight growth references (*p* = 0.003 and *p* = 0.005, respectively). Estimates from the stratified analysis of predictors were not meaningfully changed by the year of admission (Appendix A).

## 4. Discussion

The present study demonstrated a difference in the proportion of infants identified as SGA at birth on the usage of INTERGROWTH-21st growth charts compared to Fenton2013, with SGA being higher on INTERGROWTH-21st. This finding is in line with the study of Tuzun et al. which revealed that one in four cases evaluated as SGA based on the INTERGROWTH-21st curves fell within the normal range according to Fenton’s standards. Notably, these SGA infants did not have an increased risk of early morbidities [40]. On the contrary, in another study, Reddy et al. pointed out that neonates classified as SGA at birth by INTERGROWTH-21st and not by Fenton2013 growth charts had a higher incidence of morbidities such as sepsis and ROP [41]. In other studies, no significant difference was observed for the classification of birth weight for GA in preterm neonates born before 33 weeks between the two assessment growth charts [42,43]. It is very essential to accurately define SGA with appropriate charts, since being born SGA is associated with an increased risk of higher mortality, postnatal growth failure, and neurodevelopmental impairment at 18–22 months of corrected age [44]. Also, it has been shown that increasing SGA severity had a significant impact on neonatal outcomes among very preterm infants [45]. 

Our study revealed that extremely and very preterm neonates had a higher prevalence of EUGR at discharge concerning weight, according to Fenton2013 charts compared to INTERGROWTH-21st charts. This is in accordance with other studies [46,47,48], and particularly, a recent study by Starc et al. reported a decreased prevalence of EUGR of weight when INTERGROWTH-21st charts were used in comparison to Fenton. The differences in outcomes between these two growth charts were expected and can be attributed to the fundamental disparities in the way they were created. The INTERGROWTH-21st charts were developed based on the growth of preterm infants developing in an extrauterine environment, while the Fenton charts were based on the growth of fetuses developing within the confines of the intrauterine setting. The extrauterine environment presents a completely different context for preterm babies, exposing them to unique metabolic responses and morbidities, which contrast significantly with the conditions experienced by fetuses in utero. A retrospective study published in 2021 presented a lower percentage of EUGR with respect to weight and length at discharge when INTERGROWTH-21st curves were used, compared to Fenton2013 curves, and this was associated with poorer language development assessed either at 12 or 24 months [43]. It is important to note that the main limitation of the INTERGROWTH-21st preterm postnatal standards is the limited number of infants born before 33 weeks’ gestation who contributed to the development of the growth curves, which potentially affects the validity of the standards. Therefore, further research with large multicenter population-based data is needed to investigate the implications and effectiveness of utilizing these preterm postnatal growth curves, especially for monitoring growth in preterm infants at the earliest gestational ages [49]. In terms of HC, our study revealed a higher prevalence of EUGR when the INTERGROWTH-21st postnatal growth charts were used as the reference. However, other studies found no significant difference in the prevalence of EUGR concerning HC at discharge between the Fenton2013 and INTERGROWTH-21st growth charts [40,50].

Enteral feeding of preterm neonates has undergone significant advancements over time, and these progressions are evident in our study, which spanned fifteen years. Our study findings indicate a substantial decrease, after the year 2018, in the time it took to initiate enteral feeding and a notable reduction in the duration required to achieve full enteral feeding. According to recent meta-analyses, the prompt initiation of enteral feeding, preferably with mother’s own milk, combined with a faster increment in enteral feed volumes, has been associated with a shorter duration to attain full enteral feeding, decreased length of hospitalization, and a potential decrease in the occurrence of invasive infections [51]. The previous assumption that gradually advancing milk feeds at a slow rate effectively decreases the risk of necrotizing enterocolitis in very low birth weight infants has been found to be ineffective [52].

In our study, statistically significant associations were observed between a higher decrease in weight at discharge and specific factors, including delayed initiation of enteral feeding, delayed attainment of full feedings by at least one week, and a longer duration of parenteral feeding with both Fenton and INTERGROWTH-21st charts. Furthermore, based on the Fenton 2013 reference, there was a notable 7% increase in the risk of EUGR for each day of delayed enteral nutrition initiation. Since enteral feeding is insufficient, the nutrients are delivered parenterally, and prolonged parenteral nutrition in combination with long periods of interrupted enteral feeding can lead to decreased feeding tolerance and increased risk of infection, which can negatively impact growth and development in preterm infants, as demonstrated in our research findings and other studies as well [53,54]. Indeed, an early initiation of enteral feeding is important in reducing the risk of infections, improving intestinal development and maturation, stimulating microbiome development, and simultaneously promoting growth in preterm infants [55].

Our study investigated various clinical factors that are linked to EUGR using both growth references. These factors included BPD, late-onset sepsis, anemia, hsPDA and ROP. Notably, anemia and hsPDA were identified as risk factors, as they were associated with decreased weight at discharge, according to both growth charts. Premature neonates, specifically those with hsPDA, experience reduced blood flow in the superior mesenteric artery, which raises the likelihood of developing feeding problems and consequently EUGR. Additionally, the administration of nonsteroidal drugs for PDA treatment can potentially lead to gastrointestinal bleeding, causing these newborns to restrict their milk intake and delay the progression of enteral feeding [56]. Moreover, it is noteworthy that infants who experienced late-onset sepsis exhibited a threefold increased risk of developing EUGR when evaluated using the INTERGROWTH-21st growth curves. According to other studies, it was indicated that despite the lower prevalence of EUGR neonates based on weight according to INTERGROWTH-21st growth charts compared to Fenton, these neonates presented with a higher incidence of morbidities, and thus, these charts may be associated with adverse clinical outcomes during hospitalization [50,57].

Our research concluded that the prevalence of SGA and EUGR neonates differs between the Fenton2013 and INTERGROWTH-21st growth charts, reinforcing the importance of optimizing in-hospital and post-discharge nutritional support. Choosing an appropriate growth assessment tool for monitoring the postnatal growth of preterm infants is crucial to promote optimal neurodevelopmental outcomes and avoid excessive caloric intake that is linked to increased cardiometabolic risk later in life [58,59]. Prematurity constitutes an independent risk factor for the development of cardiovascular disease and metabolic syndrome regardless of birth weight [60]. Indeed, children born SGA, especially those who undergo rapid weight catch-up during early life, are prone to developing insulin resistance and central adiposity since early childhood. Furthermore, they may also experience cardiovascular dysfunctions in their adulthood [61]. Therefore, the best approach to address this barrier is implementing an evidenced-based enteral feeding protocol within each NICU and individualized care meeting each patient’s medical needs.

Our study has several limitations. As only body sizes at birth and discharge were registered in the database, information on growth patterns such as weight loss during the first few days after delivery, body sizes at specified timing (i.e., PMA 36 weeks), or other growth indicators such as growth velocity were not provided. This restraint of available data pinpoints the importance of maintaining more comprehensive and detailed records, which could be useful for future reference to the health professionals and researchers, who can utilize this data to enhance the care and treatment offered to patients. Additionally, the present study did not investigate long-term outcomes of growth and neurodevelopment. Finally, regarding dietary data, our study did not include the advancement of enteral feed volume, the type of enteral feeding (breast milk, formula feeding) and the content of parenteral nutrition in protein, lipids, and energy. 

## 5. Conclusions

In our study, despite the observed differences between the Fenton 2013 and INTERGROWTH-21st growth charts, it is not possible to definitively conclude that one graph is superior to the other. To evaluate the practical implications of the INTERGROWTH-21st growth charts specifically for very and extremely preterm neonates, additional validation using a large multicenter population-based dataset is necessary. It is crucial to closely monitor the growth of preterm infants using an appropriate growth curve to ensure optimal developmental and growth potential. Further research is warranted to deepen our understanding of the intricate relationship between nutrition and growth outcomes in preterm infants. By continuously refining dietary protocols and closely monitoring growth parameters, healthcare providers can enhance their ability to guide the nutritional management of them effectively. This ongoing effort is vital to improving the long-term health and overall well-being of these vulnerable individuals.

## Figures and Tables

**Figure 1 nutrients-15-03290-f001:**
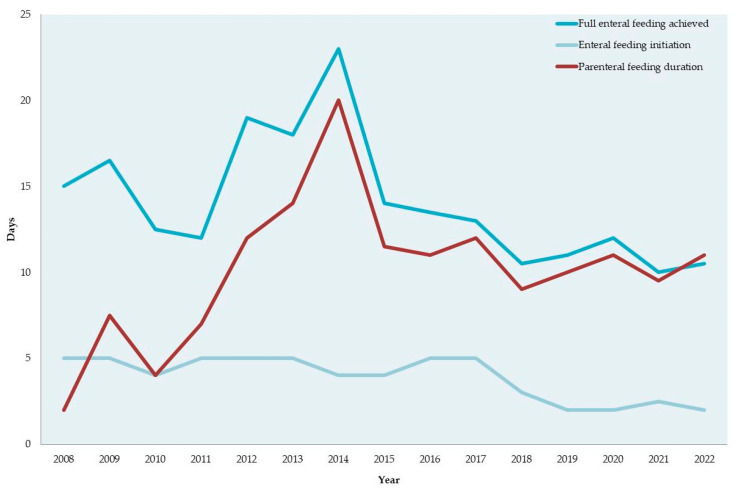
Median number of days of parenteral feeding duration, introduction of enteral feeding and full enteral feeding achieved by year of study.

**Figure 2 nutrients-15-03290-f002:**
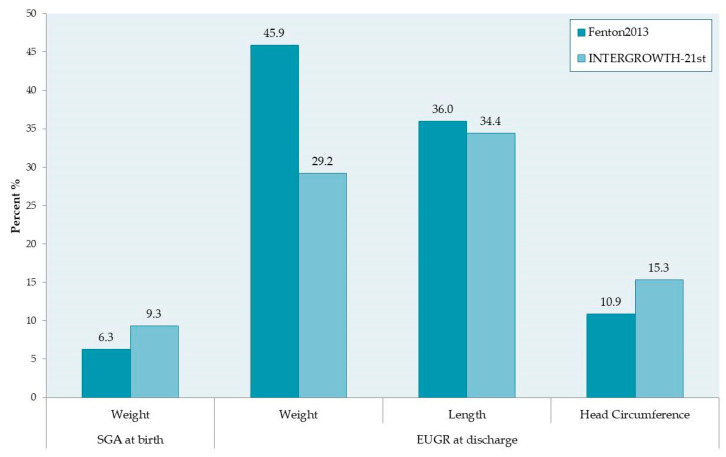
Prevalence of small for gestational age (SGA) and extrauterine growth restriction (EUGR) by Fenton2013 and INTERGROWTH-21st growth references.

**Table 1 nutrients-15-03290-t001:** Perinatal, clinical, and nutritional characteristics of the study population (*n* = 462).

	Mean ± SD or *n* (%)
Perinatal characteristics	
Gestational age (weeks)	29.6 ± 1.7
Extremely preterm (<28 weeks)	81 (17.5)
Birthweight (g)	1341.5 ± 363.3
Male sex	258 (55.8)
Caesarean section	406 (88.3)
Multiple birth	194 (43.6)
At discharge	
Postmenstrual age (weeks)	37.3 ± 2.9
Weight (g)	2415.9 ± 406.7
Length (cm)	46 ± 2.8
Head circumference (cm)	33.1 ± 1.8
Nutritional information	
Parenteral feeding duration (days)	15 ± 14.5
Enteral feeding starting time (days)	4.8 ± 4.2
Full enteral feeding achieved (days)	16.4 ± 11.3
Clinical information	
Duration of hospitalization (days)	54 ± 25.8
BPD	66 (14.3)
hsPDA	43 (9.3)
Early-onset sepsis (suspected or culture proved)	49 (10.6)
Late-onset sepsis (suspected or culture proved)	122 (26.4)
NEC	8 (1.7)
IVH (≥Grade II)	53 (11.5)
ROP (≥Stage II)	31 (6.7)
Pneumothorax	21 (4.5)
Duration of respiratory support (days)	12.9 ± 17.0
Duration of mechanical ventilation (days)	3.3 ± 7.6
Duration of non-invasive ventilation (days)	9.7 ± 13.3
Duration of oxygen therapy (days)	11.6 ± 18.3
Anemia	211 (45.7)
Cystic PVL	30 (6.5)

Abbreviations: BPD, bronchopulmonary dysplasia; hsPDA, hemodynamically significant patent ductus arteriosus; NEC, necrotizing enterocolitis; IVH, intraventricular hemorrhage; ROP, retinopathy of prematurity (≥Stage II); PVL, periventricular leukomalacia.

**Table 2 nutrients-15-03290-t002:** Infant size at birth and discharge by the Fenton2013 and INTERGROWTH-21st growth references.

	Fenton2013	INTERGROWTH-21st	*p*-Value *
At birth			
Birth weight z, mean ± SD	0.1 ± 0.9	0.1 ± 1.1	0.001
SGA weight, n (%)	29 (6.3)	43 (9.3)	<0.001
At discharge			
Weight z score, mean ± SD	−1.2 ± 1.4	−0.7 ± 1.5	<0.001
Length z score, mean ± SD	−0.9 ± 1.5	−0.7 ± 2.0	<0.001
HC z score, mean ± SD	−0.1 ± 1.2	0.0 ± 1.6	<0.001
EUGR weight, n (%)	209 (45.9)	133 (29.2)	<0.001
EUGR length, n (%)	165 (36.0)	158 (34.4)	0.039
EUGR HC, n (%)	50 (10.9)	70 (15.3)	<0.001

Abbreviations: SGA, small for gestational age; HC, head circumference; EUGR, extrauterine growth restriction. * *p*-values from paired *t*-tests were used to compare sex-specific z-scores at birth and discharge, and Cochran’s Q test was used to compare the prevalence of SGA and EUGR between growth references (Fenton2013 and INTERGROWTH-21st).

**Table 3 nutrients-15-03290-t003:** Perinatal, infant nutrition and clinical factors ^1^ by extrauterine restricted growth (EUGR) based on weight at discharge by the Fenton2013 and INTERGROWTH-21st growth references.

	Fenton2013			INTERGROWTH-21st		
	Restricted	Non Restricted	*p*-Value ^2^	Restricted	Non Restricted	*p*-Value ^2^
Perinatal factors						
Male sex	111 (53.1)	141 (57.3)	0.395	81 (60.9)	171 (53.1)	0.147
Gestational age (weeks)	29.2 ± 1.8	30.0 ± 1.5	<0.001	28.9 ± 1.9	29.9 ± 1.5	<0.001
Extremely preterm (<28 weeks)	48 (23.0)	30 (12.2)	0.003	38 (28.6)	40 (12.4)	<0.001
Birth weight (g)	1144.3 ± 288.1	1510.9 ± 334.6	<0.001	1064.2 ± 276.5	1457.5 ± 331.3	<0.001
BW z-score	−0.3 ± 0.8	0.5 ± 0.9	<0.001	−0.7 ± 1.4	0.4 ± 0.8	<0.001
SGA	26 (12.4)	3 (1.2)	<0.001	35 (26.3)	8 (2.5)	<0.001
Caesarean section	186 (89.0)	213 (87.3)	0.663	116 (87.2)	283 (88.4)	0.751
Multiple birth	114 (56.4)	131 (55.5)	0.848	78 (60.0)	167 (54.2)	0.293
Duration of hospitalization (days)	66.3 ± 27.5	43.5 ± 19.1	<0.001	75.1 ± 29.4	45.2 ± 18.2	<0.001
Nutritional factors						
Parenteral feeding duration (days)	11.5 (6–24)	8 (4–13)	<0.001	14 (8–30)	8 (4–13.5)	<0.001
Enteral feeding initiation (days)	4 (2–8)	3 (2–5)	<0.001	5 (3–9)	3 (2–5)	<0.001
Full enteral feeding achieved (days)	18 (11–27)	11 (8–16)	<0.001	19 (12–30)	11 (8–18)	<0.001
Clinical factors						
BPD	38 (18.2)	28 (11.4)	0.045	29 (21.8)	37 (11.5)	0.008
Early-onset sepsis	29 (13.9)	20 (8.1)	0.068	16 (12.0)	33 (10.3)	0.619
Late-onset sepsis	74 (35.4)	46 (18.7)	<0.001	64 (48.1)	56 (17.4)	<0.001
Anemia	117 (56.0)	92 (37.4)	<0.001	90 (67.7)	119 (37.0)	<0.001
NEC	8 (3.8)	0 (0.0)	0.002	7 (5.3)	1 (0.3)	0.001
hsPDA	31 (14.8)	11 (4.5)	<0.001	25 (18.8)	17 (5.3)	<0.001
ROP	22 (10.5)	9 (3.7)	0.005	17 (12.8)	14 (4.4)	0.002
Respiratory support (days)	17.0 ± 20.5	9.5 ± 12.5	<0.001	20.7 ± 21.1	9.7 ± 13.9	<0.001
Mechanical ventilation (days)	5.5 ± 10.4	1.5 ± 2.9	<0.001	7.0 ± 11.6	1.8 ± 4.4	<0.001
Non-invasive ventilation(days)	11.7 ± 14.8	8.0 ± 11.5	0.003	13.9 ± 15.3	8.0 ± 11.9	<0.001
Oxygen administration (days)	16.8 ± 23.0	7.2 ± 11.7	<0.001	21.0 ± 24.9	7.7 ± 13.2	<0.001
Cystic PVL	18 (8.6)	12 (4.9)	0.130	13 (9.8)	17 (5.3)	0.096
IVH (Grade ≥ II)	11 (5.3)	12 (4.9)	0.999	6 (4.5)	17 (5.3)	0.818

Abbreviations: BPD, bronchopulmonary dysplasia; BW, birth weight; hsPDA, hemodynamically significant patent ductus arteriosus; IVH, intraventricular hemorrhage (grade ≥ II); NEC, necrotizing enterocolitis; ROP, retinopathy of prematurity (≥Stage II); cystic PVL, cystic periventricular leukomalacia. ^1^ Numbers represent mean ± SD for parametric continuous variables, median (IQR) for non-parametric continuous variables and *n* (%) for categorical variables. ^2^ *p*-values obtained from independent samples *t*-test or Mann–Whitney non-parametric test, used to compare continuous variables between two groups, and Fisher’s exact test, used to compare categorical variables.

**Table 4 nutrients-15-03290-t004:** Associations * of clinical and nutritional factors with extrauterine growth restriction based on the Fenton2013 and INTERGROWTH-21st weight growth references.

	Fenton2013		INTERGROWTH-21st	
	Weight z-Score	EUGR	Weight z-Score	EUGR
	β (95% CI)	OR (95% CI)	β (95% CI)	OR (95% CI)
Perinatal factors				
Extremely preterm	0.53 (0.23, 0.84)	0.49 (0.19, 1.27)	0.48 (0.17, 0.80)	0.67 (0.26, 1.76)
BW z-score	0.36 (0.26, 0.45)	0.21 (0.14, 0.31)	0.43 (0.33, 0.53)	0.27 (0.19, 0.39)
Hospitalization (days)	−0.04 (−0.04, −0.03)	1.05 (1.03, 1.07)	−0.04 (−0.04, −0.03)	1.05 (1.03, 1.08)
Nutritional factors				
PN duration (days)	0.01 (0.00, 0.02)	0.97 (0.94, 1.01)	0.01 (−0.00, 0.02)	0.99 (0.95, 1.03)
EN initiation (days)	−0.02 (−0.05, 0.00)	1.04 (0.94, 1.14)	−0.03 (−0.06, 0.00)	1.16 (1.05, 1.28)
Full EN achieved (days)	−0.02 (−0.03, −0.01)	1.07 (1.03, 1.12)	−0.02 (−0.04, −0.01)	1.03 (0.99, 1.07)
Clinical factors				
BPD	0.28 (−0.07, 0.63)	0.81 (0.27, 2.40)	0.37 (0.00, 0.73)	0.98 (0.33, 2.97)
Late-onset sepsis	−0.07 (−0.30, 0.15)	1.33 (0.70, 2.52)	−0.13 (−0.37, 0.10)	3.08 (1.56, 6.08)
Anemia	0.33 (0.10, 0.56)	0.49 (0.25, 0.95)	0.31 (0.06, 0.55)	0.55 (0.25, 1.19)
hsPDA	−0.41 (−0.75, −0.07)	2.59 (0.88, 7.61)	−0.39 (−0.75, −0.04)	1.67 (0.56, 4.95)
ROP	0.14 (−0.29, 0.57)	2.04 (0.52, 8.07)	0.08 (−0.36, 0.53)	0.99 (0.25, 3.86)
Respiratory support (days)	0.00 (−0.01, 0.01)	0.99 (0.95, 1.02)	0.00 (−0.01, 0.01)	1.00 (0.97, 1.03)
Oxygen administration (days)	0.00 (−0.00, 0.01)	1.00 (0.97, 1.03)	0.00 (−0.01, 0.01)	0.99 (0.96, 1.01)

Abbreviations: SGA, small for gestational age; PN, parenteral nutrition; EN, enteral nutrition; BPD, bronchopulmonary dysplasia; hsPDA, hemodynamically significant patent ductus arteriosus; ROP, retinopathy of prematurity (≥Stage II); cystic PVL, cystic periventricular leukomalacia. * Beta coefficients (β) and odds ratios (OR) and corresponding confidence intervals (CI) obtained using linear and logistic regression models, respectively.

## Data Availability

The data presented in this study are available on request from the corresponding author. The data are not publicly available due to privacy or ethical restrictions.

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
