# Peer review of "The Prevalence of Small for Gestational Age and Extrauterine Growth Restriction among Extremely and Very Preterm Neonates, Using Different Growth Curves, and Its Association with Clinical and Nutritional Factors"

_nutrients, 2023, doi:10.3390/nu15153290_

Round 1

Reviewer 1 Report

There was little that was novel in this article. The nutritional and clinical factors that played into EUGR are well established. As noted in the discussion, the Intergrowth-21st growth curves are primarily based on older GA infants, and thus makes it difficult to extrapolate the validity of this curve in measuring more immature and smaller infants. 

The change in nutritional feeding practices during the course of this retrospective study greatly changed the number of days of parenteral feedings and time to full enteral feedings. Both of these factors are known to contribute to higher rates of EUGR. Providing an overall percentage of EUGR for the entire time frame is not very useful - it was likely higher earlier and lower later. The data would probably be more accurate and applicable if it was divided into two different epochs, though it would decrease the likelihood of significant results. 

Similarly, weight at discharge can be highly variable, with a span of almost 6 weeks covered. Rate of EUGR at a defined time point (ie 36 weeks) might be a more relevant data point. 

Would like to see more information in the discussion about what the authors think the rationale is for the differences in rates of EUGR for the various growth parameters between Fenton and Intergrowth. This seemed to have been the main reason for the study, but there was little delving into the topic.

Reviewer 2 Report

Manuscript titled “The prevalence of SGA and EUGR among preterm neonates born before 32 weeks of gestation, using different growth curves, and its association with clinical and nutritional factors” reports an analysis that compares the use of two different tools to monitor neonates’ growth, Fenton 2013 and INTERGROWTH-21st. The study discusses differences in using these tools, and their implications, moreover, the authors acknowledge limitations of their study, such as lack of various pieces of information that were not available for them to take into account in their analysis. The introduction provides an adequate background that explains the authors’ reasoning and justification to undertake their work, materials and methods are adequately described, results are properly presented, discussion is adequate and the conclusion is congruent with the data presented on previous sections. The work is interesting, since its contrast between different instruments highlights the utility and application of each one. There are some minor comments for the authors:

1. Please consider writing out in full the terms “SGA” and “EUGR” on your manuscript’s title instead of abbreviating them.

2. On Table 1, please homogenize the number of decimals given for a variable and its error. For example, birthweight is mentioned as “1341.5 +- 363.33”; please homogenize both to either one or two decimals. Also in this table, please mention “g” instead of “grams”.

3. Please fix a typo on the Y axis of Figure 2 (“Pecrcent”).

4. In line 272, “this was associated with poorer language development”. Please specify at what age was this finding documented (if this is available).

5. The final paragraph of the discussion mentions a lack of various pieces of information that the authors consider limitations of their study. This observation also highlights the need to maintain more detailed records, since this information could be useful not only for the patients themselves, but for clinicians or researchers who may use these data to improve the care provided to future patients.
